# The Lipidomics of Spermatozoa and Red Blood Cells Membrane Profile of Martina Franca Donkey: Preliminary Evaluation

**DOI:** 10.3390/ani13010008

**Published:** 2022-12-20

**Authors:** Paraskevi Prasinou, Ippolito De Amicis, Isa Fusaro, Roberta Bucci, Damiano Cavallini, Salvatore Parrillo, Maurizio Caputo, Alessandro Gramenzi, Augusto Carluccio

**Affiliations:** 1Faculty of Veterinary Medicine, University of Teramo, Piano d’Accio, 64100 Teramo, Italy; 2Department of Veterinary Medical Science, University of Bologna, 40064 Bologna, Italy

**Keywords:** donkey, equine lipidomic profile, red blood cell membrane, spermatozoa, fatty acids, progressive motility, working equids

## Abstract

**Simple Summary:**

Both in human medicine and in the veterinary field, lipidomic techniques have been applied extensively in order to evaluate health status. In equine species where the subject of focus is the fertility and sperm quality of the equids, lipidomic assessment can be a useful tool to better understand the metabolic transformation of the fatty acids of the spermatozoa as well as the erythrocyte membrane and their role on spermiogenesis. The purpose of this study is to evaluate the composition of the spermatozoa’s membranes of 26 healthy male Martina Franca donkeys and the possible correlation with the sperm parameters and the erythrocytes’ membrane. The results demonstrate that the membrane of erythrocytes and spermatozoa are different, which provides a deeper understanding of the distribution and the role of these fatty acids in sperm parameters that are highly associated with fertility.

**Abstract:**

Fatty acid-based lipidomic analysis has been widely used to evaluate health status in human medicine as well as in the veterinary field. In equine species, there has been a developing interest in fertility and sperm quality. Fatty acids, being the principal components of the membranes, play an active role in the regulation of the metabolic activities, and their role on spermiogenesis seems to be of great importance for the resulting quality of the sperm and, thus, fertility. With the application of widely used lipidomic techniques, the aim of this study was to evaluate: (a) the fatty acid content of the spermatozoa’s membranes of 26 healthy male Martina Franca donkeys and its possible correlation with sperm parameters, and (b) the evaluation of the composition of the red blood cells’ membrane. PUFA omega-6 are the principal components (40.38%) of the total PUFA content (47.79%) in both types of cells; however, DPA is the predominant one on the spermatozoa’s membrane (27.57%) but is not present in the erythrocyte’s membrane. Spermatozoa’s motility (%) is positively correlated with stearic acid and EPA, and progressive motility (%), with oleic acid. These findings offer information on the composition of both types of cells’ membranes in healthy male MF donkeys and reflect the metabolic transformations of the spermatozoa’s membrane during the maturation period, providing a better perception of the role of fatty acids in sperm parameters and fertility.

## 1. Introduction

Membrane phospholipids and their saturated (SFAs), monounsaturated (MUFAs) and polyunsaturated fatty acids (PUFAs) as hydrophobic components have fundamental structural and functional roles in cell life and homeostatic balance [1,2,3]. The study of all lipid classes, the interaction of a cell with its environment, as well as the communication between different cells with one another, compose the principle of lipidomics [4]. Over the past few decades, lipidomic techniques have been widely used to gather information about numerous lipids and their fatty acids, i.e., the metabolic pathways of a living organism. Moreover, the membrane lipidome profile of erythrocytes has a meaning for the metabolic and nutritional status of living organisms, providing information on health conditions when the required lipidome composition is balanced [1]. That means that membrane phospholipids containing fatty acids as hydrophobic components with important structural and functional roles for cell life must express the content of saturated, monounsaturated and polyunsaturated moieties with a specific combination to provide homeostatic balance. In animal studies, the membrane lipidome composition of experimental models has been thoroughly studied in relation with dietary regimes and has showed the importance of PUFA balance for the membrane properties [5].

It has been demonstrated that fatty acids play a crucial role on the health status of all living organisms, both human and animals [6], with the erythrocyte membrane fatty acid content being the main subject of research in animals through the last years [7,8]. A previous study showed that an altered lipid metabolism can result in the development of metabolic syndrome in equids [9]. However, the veterinary field (in particular the equine sector of veterinary research) is still short on systematic studies that can provide information about lipid metabolism. Most importantly, there is a complete lack of information on the composition of erythrocytes’ membranes.

In addition, while there is an increasing interest on the sperm parameters and the sexual behavior of working equids [10], scarce and imprecise data are reported on the composition of spermatozoa’s membrane. Only a few studies have been conducted on equine spermatozoa lipidome or erythrocyte membranes to provide new information about lipid homeostasis [11,12]. Nolazco Sassot et al., 2018, conducting a study on Thoroughbreds, indicate changes in specific plasma lipids through supramaximal exercise that could be useful to formulate a specific equine diet [13]. In this regard, the role of micronutrients was widely explored in prevention of fatty acid oxidation in all the membranes of the cellules [14]. There are only a few studies focusing on fatty acids, but they seem to be limited by either including a small number of samples or not always reporting the same cohort of fatty acids, focusing only on PUFAs (ω3 and ω6) and not considering the rest of the lipid classes [15]. A possible reason for this could be what seems to be a positive correlation between fertility and high levels of PUFAs on the spermatozoa membrane [16]. Previous studies demonstrate that PUFAs are found in higher concentrations in the membrane spermatozoa compared to the others fatty acids families, and ω6 are more prevalent than ω3, with DPA being the predominant ω6 fatty acid [11,17]. However, while there is a growing interest in the lipidomic assessment of spermatozoa’s membranes to assess PUFAs, to this point, there is a lack of systematic studies that provide information about the total composition of the spermatozoa’s membrane, including fatty acids that are representative of all lipid classes, as well as its possible correlation with animal characteristics and sperm characteristics in equine species. Animal characteristics such as age and BMI could be a possible factor of influence for sperm quality [18,19,20]. The composition of the spermatozoa of the membrane can be an important indicator of semen quality, and, in turn, semen quality is an important indicator of male reproductive health and fertility [16,21]. Therefore, the need to study the distribution of these fatty acids into the different parts of the spermatozoa is crucial, since it can provide important information about their motility.

The aim of this study is: (a) the evaluation of the fatty acid (FA) composition of the spermatozoa’s membrane phospholipids (PL) in healthy Martina Franca (MF) donkeys (total of 26 subjects) and the correlation between the composition of the spermatozoa’s membrane FA and the sperm characteristics; and (b) the evaluation, for the first time, of the FA of the red blood cell’s (RBC) PL in the same donkeys. The general objective of this work is to evaluate the correspondence between fatty acid-based lipidomic analysis and sperm traits in donkeys of the Martina Franca breed.

## 2. Materials and Methods

### 2.1. Inclusion Criteria and Sample Collection

Semen as well as blood samples were collected from a population of twenty-six healthy male donkeys (Martina Franca), with age between 2 and 19 years. The Martina Franca donkey (MF) has been considered to be an endangered breed since the early 1980s [22,23], and its population consists of less than 200 females and 40 males [24].

The samples were collected during the summer season, from June to August 2021. Each of the stallions that we selected for this research project had been approved for assisted reproduction and had been involved in breed-mating programs. Thereby, the donkeys underwent a training protocol for semen collection procedures. For each subject, two days before the samples collection, BCS [25,26] and health conditions were assessed by veterinary physical inspection and were recorded (temperature, heart and respiratory rate, auscultation of the heart, lungs and gut). The detailed characteristics of the donkeys involved in the study are presented in Appendix A. Sperm as well as blood samples were collected by staff specialists working at the Veterinary Medicine Department of Teramo. Both semen collection and sperm analysis conducted for the realization of this project were also essential aspects of the regular andrological visit. A Missouri artificial vagina was used for the collection of the semen samples. Before the start of the trial, all animals were subjected to semen collection every 3 days for 1 week, and the material was discarded to eliminate extra gonadal reserves. Then, each stallion underwent semen collection once every two weeks up to a total of three times during the trial.

Blood samples (1 mL each) were collected in Ethylenediaminetetraacetic acid as tripotassium salt (EDTA) tubes.

A specific diet that consisted of hay (about 2% of live weight) with a supplementation of 1.5 kg of commercial concentrate feeding was administered two times per day (9 a.m. and 3 p.m.). The hay quality, adequacy and palatability were checked according to [27,28]. Water was provided *ad libitum* in linear drinking troughs. For the duration of this trial, the hay samples and concentrate were collected every two weeks and analyzed for dry matter content (DM), ash and crude protein (CP) following the official protocols of the AOAC (1990). Neutral detergent fiber (NDF), acid detergent fiber (ADF) and lignin (ADL) were analyzed as previously described [29]. The data of the feed analysis are reported in Table 1.

### 2.2. Ethical Statement

The project has been approved by the Ethic Committee of the Department of Veterinary Medicine of the University of Study of Teramo, Prot. n. 18532 of 28 June 2022.

### 2.3. Sperm Assay/Spermiogram (Analysis of Seminal Fluid)

Semen was collected in a volume-marked vessel, then filtered in another prewarmed similar vessel to assess total volume and gel-free volume. The semen assessment was performed as close to the time of collection as possible. Concentration and viability were assessed with NucleoCounter SP-100 (chemometec). Sperm motility (total and progressive) was evaluated with a CASA System (IVOS II, Hamiltorn Thorne, IMV Tech. Beverly, MA, USA)

### 2.4. Lipidomic Profile

For spermatozoa lipidomic profile, using a whole sperm sample of known concentration, the sperm cells were isolated from the seminal plasma by two consecutive centrifugations (2500 rpm × 10 min each) followed by seminal plasma removal, and the cells were washed with PBS (0.5 mL, pH = 7.8) two times (2500 rpm × 5 min each). The sperm cells were resuspended in pure water (18 mQ), obtaining a concentration of 6 × 10^6^ spermatozoa per 1 mL, which was used to isolate membrane lipids using 2:1 chloroform: methanol as the organic phase [30].

The organic layer was extracted and the sample was brought under vacuum to dryness. In order to determine the efficacy of the lipid extraction, a thin layer chromatography (TLC) using chloroform/methanol/water 65:25:4 was used [31]. The phospholipid extract was transesterified at room temperature for 10 min with 0.5 M KOH/MeOH to obtain the corresponding methyl esters (FAMEs). This chemical transformation was carried out with known procedures, checking for absence of oxidative and degradation reactions, which could affect the final fatty acid composition. For each sperm sample, the analysis was repeated two times and, in addition, the analysis of the same sample was performed twice.

For erythrocyte lipidomic profile, the erythrocytes were isolated from whole blood in EDTA and underwent membrane lipidome analysis, as previously described [7]. Briefly, starting from a 1 mL whole blood sample, the separation of erythrocytes from plasma followed by washings and centrifugation for the membrane isolation finally gave a pellet that was re-suspended in pure water and used to extract PL lipids using 2:1 chloroform: methanol as the organic phase following the same steps described above. The detailed list of materials used is reported in Appendix A.

The final steps were the extraction of the FAMEs using n-hexane, evaporation under vacuum to dryness and use of the FAME extract to perform gas chromatography (GC) analysis, as described below.

### 2.5. GC Analysis of FAMEs

First, GC analysis of the commercially available reference standard materials for each of the selected 12 fatty acids was performed as described in Appendix A. Calibration curves were obtained for the quantitative analysis of each peak of the chromatogram and are shown in Appendix A.

The FAME mixture obtained from the sperm as well as the erythrocyte membrane pellet was dissolved in 20 μL of n-hexane, and 1μL was directly injected to the Agilent 7890B GC system equipped with a flame ionization detector and a DB-23 (50%-Cyanopropyl)-methylpolysiloxane capillary column (60 m, 0.25 mm i.d., 0.25 μm film thickness). The initial temperature was 165 °C, held for 3 min, followed by an increase of 1 °C/min up to 195 °C, held for 40 min, followed by a second increase of 10 °C/min up to 240 °C, held for 10 min. The carrier gas was hydrogen, held at a constant pressure of 16.482 psi. All the FAMEs were identified by comparison with the retention times of standard references either commercially available or obtained by synthesis.

### 2.6. Evaluation of the Fatty Acid Cluster, Corresponding Families and Homeostasis Indexes

A selected group of 12 fatty acids that are representative of the main fatty acid families present in sperm membrane was chosen. It consisted of myristic (C14:0), palmitic (C16:0) and stearic (C18:0) acids as SFAs; palmitoleic (9c,C16:1), oleic (9c,C18:1) and vaccenic (11c,C18:1) acids as MUFAs; linoleic (LA, C18:2), dihomo-gamma-linolenic (DGLA;C20:3), arachidonic (AA, C20:4) and docosapentaenoic (DPA;C22:5) acids as PUFAs ω-6; and eicosapentaenoic (EPA, C20:5) and docosahexaenoic (DHA, C22:6) acids as PUFAs ω-3. Using the calibration curves, the quantitative values for each peak were obtained as μg/mL, which allowed to also calculate the total fatty acid contents (total SFAs, total MUFAs, total PUFAs), the ratios between the families (SFA/MUFA, omega-6/omega-3) and three indexes (unsaturation index (UI), peroxidation index (PI), PUFA balance). The gas chromatographic method was able to separate satisfactorily all the 12 fatty acids without superimposition of other peaks, in particular the positional and geometrical isomers of the unsaturated fatty acids, using conditions that have been experienced in previously reported papers [32,33,34].

The group of 12 fatty acids corresponds to chromatographic peak areas >97% were dected for sperm and for red blood cells. The quantification of the fatty acids was executed by known calibration procedures, as reported in Appendix A. 

The data obtained from the sperm cells lipidomic analysis were then used to evaluate the sperm parameters obtained from the spermiogram (reaction time (min), total volume (mL), concentration (mln/mL), dead (mln/mL), motility %, progressive motility %). For the evaluation of progressive motility, the donkeys were divided into three groups [35]: those who demonstrated spermatozoa with a high progressive motility that was determined equal or above 45% [21]; those with a low progressive motility (below 35%); and those that demonstrated spermatozoa with a medium progressive motility (35% ≤ medium PM < 45%), following a previously applied model [35].

### 2.7. Statistical Methods

Statistical analysis of the obtained data was performed with the GraphPad Prism 6.01 software (GraphPad Software, Inc., San Diego, CA, USA) and JMP Pro 16 (SAS Institute Inc., Cary, NC, USA). All data were evaluated using a standard descriptive statistic and reported as mean ± SD or as median and range (minimum–maximum), based on their distribution. Normality was checked graphically or using the D’Agostino–Pearson test. A comparison between groups was drawn using the mixed-model procedure, where each donkey was included as an experimental unit and each sampling as a repeated measure. After the analysis, residuals were rechecked for normality and, when necessary, parameters were BoxCox transformed and the analysis was repeated. A regression analysis (Spearman) was used to evaluate the correlation between fatty acid percentages with the donkey’s characteristics and the sperm parameters. The threshold for the statistical significance (*p*-value) was set up at 0.05. Only the results with a significant correlation are presented below. The distribution graphs were produced with the Past 3.14 software (free download; Øyvind Hammer, Natural History Museum, University of Oslo, Oslo, Norway).

## 3. Results

### 3.1. Semen Analysis of Healthy Donkeys

The results obtained from the semen analysis (the donkeys’ ID, the reaction time (min), the total volume (mL), the gel-free volume (mL), the concentration (mln/mL), the number of dead spermatozoa (mln/mL), the motility (%) and the progressive motility (%)) are presented in the following table (Table 2). The individual values of all the sperm parameters of each donkey are reported in Appendix A.

### 3.2. The Spermatozoa Membrane Lipidome in Healthy Donkeys

The values of single fatty acids, their respective families (total SFAs, total MUFAs, and total PUFAs), the ratio between the families (SFA/MUFA, ω-6/ω-3) and the lipid indexes (UI, PI and PUFA balance) of the spermatozoa’s membranes of the healthy donkeys are reported as median, minimum value (min), maximum value (max), mean and SD in Table 3. The distribution of each fatty acid as well as the corresponding families are presented in Appendix A.

The obtained data indicate that the PUFA content is the prevalent family (48.0%) in the donkeys’ spermatozoa membrane lipidome. PUFAs omega-6 are the principal components of the total PUFA content (40.4%), with DPA being the predominant one (27.6%), followed by arachidonic (6.6%), linoleic (4.7%) and gammalinolenic (2.0%) acids. Within the PUFA-ω3 (7.7%), DHA is the prevalent one (7.6%), followed by EPA (0.1%). Total SFAs are also found in high concentration (45.1%), with palmitic acid detected in higher levels (30.3%), followed by stearic (10.8%) and myristic (4.1%). Total MUFAs are detected in lower levels (6.8%), with vaccenic and oleic acids found in similar concentrations (3.475% and 3.2%, respectively) and palmitoleic detected in very low levels (0.1%). The SFA/MUFA ratio is relatively high (6.7) due to the lower concentrations of MUFAs. The ω-6/ω-3 ratio was relatively low (5.5), and the PUFA balance, high (16.0), which demonstrates the relatively high levels of PUFAs omega-3. The unsaturation and peroxidation indexes are both extremely high (229.9% and 258.9%, respectively). In Figure 1, the distribution graphics (% rel. quant.) for each family (total SFAs, total MUFAs, PUFAs omega-3, PUFAs omega-6 and total PUFAs) are presented.

### 3.3. Correlations of Spermatozoa FA with the Donkeys’ Characteristics

The 26 donkeys that were selected for this study had an age between 2 and 19 years (median 5 years) and a BCS between 3/5 and 4/5 (median 3/5). We used the data obtained from the lipidomic analysis to evaluate possible correlations between FAME types, families and lipid indexes, and the donkey’s characteristics that are reported in Appendix A. From the correlation between the values obtained and the BCS, no significant results were obtained.

Regarding age, the only two significant negative correlations found were with the DHA (*p* = 0.031, *r* = −0.423) and the total omega-3 content (*p* = 0.031, *r* = −0.424) as depicted in Figure 1. All the correlation graphs between the values obtained and the donkey’s age are shown in Appendix A.

### 3.4. Correlations of Spermatozoa FA with Sperm Parameters

Using the obtained data from the lipidomic analysis (Table 3), we evaluated the correlations between FAME types and the sperm parameters. From the correlation with the reaction time (min), total volume (mL), concentration (mln/mL) and dead (mln/mL), no significant results were obtained.

#### 3.4.1. Correlation of Spermatozoa FA with Motility (%)

The data obtained also show that there was a positive correlation between stearic acid (C18:0; *p* = 0.392, *r* = −0.048) and EPA (*p* = 0.514, *r* = 0.07), and motility (%), as depicted in Figure 2. All the correlation graphs between the values obtained from the spermatozoa’s membranes and the motility (%) are shown in Appendix A.

#### 3.4.2. Correlations of Spermatozoa FA with Progressive Motility (%) (PM)

The fatty acid composition of fresh donkey spermatozoa with low, medium and high PM was examined. No significant differences on the total SFA and total PUFA contents were detected among the three different groups. The levels of oleic acid [C18:1, 9c] were significantly higher on the spermatozoa with a high PM when compared to the low PM (Figure 3). However, no differences appeared on the total MUFA content between the two groups. All the correlation graphs between the values obtained from the spermatozoa’s membranes and the 3 different groups of PM are shown in Appendix A.

### 3.5. The Erythrocyte Membrane Lipidome in Healthy Donkeys

For the lipidomic assay of the erythrocytes’ membranes, we chose the same cluster of 12 fatty acids. The values of single fatty acids, their corresponding families (total SFAs, total MUFAs and total PUFAs), the ratio between the families (SFA/MUFA, ω − 6/ω − 3) and the indexes (UI, PI and PUFA balance) of the healthy donkeys are reported as mean and SD in Table 4.

The obtained data indicate that in the donkeys’ erythrocyte membrane lipidome, the PUFA content is also the prevalent family (48.0%). The PUFAs omega-6 are the principal components of the total PUFA content (47.8%), with linoleic acid being the predominant one (46.3%), while both arachidonic and gammalinolenic acids were detected in very low concentrations (1.4% and 0.1%, respectively). In erythrocytes, DPA was nonexistent. The total PUFA-ω3 were in very low levels (0.2%), with EPA being higher that DHA (0.2% and 0.03%, respectively). In contrast to the results obtained from the sperm cells, in the erythrocytes, the total MUFAs were the second family to be found in relatively high levels, with oleic acid being the most prevalent one (30.3%), and palmitoleic and vaccenic acids being detected in low concentrations (1.1% and 0.9%, respectively). Total SFAs were found in minor levels (19.7%), with palmitic and stearic acids having similar levels (10.3% and 9.4%, respectively), and myristic acid not being present. In the erythrocytes, the SFA/MUFA ratio was very low (0.6%) due to the high concentrations of MUFAs. The ω-6/ ω-3 ratio gives a very high value (222.6%), which demonstrates the prevalence of ω-6 over ω-3 PUFAs. The PUFA balance is low (0.5%), as expected due to the very low levels of both EPA and DHA. The unsaturation and peroxidation indexes are both relatively low (131.9% and 54.2%, respectively).

## 4. Discussion

For the first time, the spermatozoa as well as the red blood cell lipidome characterization of the fatty acid content of membrane’s glycerophospholipids is provided in MF donkeys. The information provided in this study is of great importance, since MF is one of the currently endangered breeds in Italy. Our cluster of choice consisted of 12 fatty acids that are representative of the three different fatty acid families (i.e., saturated, monounsaturated and polyunsaturated). Moreover, we evaluated the fatty acid cohort obtained from the erythrocyte membrane, isolated from the same donkeys (a study occurring for the first time in donkeys), with the goal of detecting a possible correlation between the lipidomes of the two different types of cells. We then performed a correlation of the obtained data and the sperm characteristics obtained from the sperm assay. Some interesting significant changes were detected between the sperm characteristics and the lipidomic membrane profile of sperm cells; however, not in the erythrocytes. A previous study conducted in humans demonstrated that sperm and erythrocytes use fatty acids differently, demonstrating how each tissue metabolizes differently the fatty acids coming from a fat supplementation [36].

As mentioned before, it is important to highlight that in the previous studies on the fatty acids of spermatozoa’s membranes in both humans and animals reported in the literature, the chosen cohort of fatty acids is not always the same [15].

The data shown in Table 2 were considered normal for the species as observed in a previous study [37].

The overall fatty acid distribution on the spermatozoa’s membranes that is reported in Table 3 shows that, in the spermatozoa’s membrane lipidome of the donkey, the PUFA content is prevalent, followed by SFAs and MUFAs. The data obtained from the erythrocytes’ membranes lipidomic analysis show that, within the distribution of the fatty acid (Table 4), the PUFA content is prevalent (as it was for the spermatozoa); however, in this case, MUFAs were found in higher levels than SFAs, demonstrating that the proportions of said families were different on the erythrocytes than the spermatozoa. Comparing our data on the FA families with the studies conducted on humans [33] or other animal species [7], it can be safely assumed that donkeys have a different erythrocyte membrane lipidome than other, different animal species; thus, this study could be the initial point of a further investigation to a deeper understanding of the metabolic pathways on the equine species.

In the spermatozoa, total SFAs are found in a high concentration, with palmitic acid being the predominant one, a result that is consistent with the literature [38]. However, in the erythrocytes, SFAs are detected in minor levels. On the other hand, and as reported in previous studies [39,40], the total MUFA content was found in lower levels in the spermatozoa when compared to the red blood cells. As a result, the SFA/MUFA ratio is relatively high on the spermatozoa, while very low in the erythrocytes.

As previously described, the total PUFA content is prevalent for both sperm cells and erythrocytes. Throughout the years, a high proportion of PUFAs has been detected in mammalian spermatozoa, and their levels influence sperm maturation, motility and acrosome reaction [41]. Considering the whole period of time, from non-breeding season all the way to breeding season, PUFA levels are higher in male mammals [12].

For the spermatozoa as well as the red blood cells, PUFAs omega-6 are the principal components of the total PUFA content. Interestingly enough, as in a lot of studies that have been conducted in the past [12,17], DPA is the prevalent ω-6 in the spermatozoa. However, in the erythrocytes’ membranes, it is nonexistent, while LA is the one found in abundance. Within the PUFA-ω3, DHA is the prevalent one in the sperm cells, as reported in previous studies [42], while in the erythrocytes, EPA is higher than DHA. The rest of the PUFA-ω6 and PUFA-ω3 are found in very small proportions for both types of cells. These results could be due to the fact that, through the different alternative metabolic pathways and through elongation and/or saturation processes, the testicular cell has the ability to convert the dietary essential FAs (specifically LA and ALA) to their possible derivatives ARA, EPA, DPA and DHA [43], a process similar to the one that occurs in the liver. The red blood cell is considered the representative cell within an organism to be studied for lipidomic analysis, since it has been demonstrated that, between different tissues of the same organism, the individual fatty acids may differ, but the families are found in similar proportions [37]. Looking at our data on both the sperm and red blood cell lipidomes, it could be hypothesized that during the process of the maturation of the sperm cells that occurs in the epididymis, the majority of the LA content that is present on the somatic cells undergoes metabolic procedures and is finally converted into its derivatives, with the main transformation being the one resulting in DPA, followed by ARA. This is a conclusion that seems to only apply to somatic cells, since the only role of germ cells is to be used to produce new organisms, something that is depicted in those differences in our data between the two different types of cells. Spermatozoa have different demands on fatty acid content and so their lipidic content is regulated locally within the male reproductive tract [44]. The maturation of spermatozoa has been thoroughly studied in animal models. These studies show that the membrane structure of the spermatozoa as it passes through the epididymis undergoes a total rearrangement of fatty acids occurs [38,45]. Essential fatty acids can be elongated and/or desaturated by the germ cell line, and there is a correlation between the levels of long-chain PUFAs and the normal morphology of sperm cells [38], which has an impact on the fertility of the males [46]. In particular, previous studies demonstrate that during the process of maturation of the spermatozoa that occurs in the epididymis the assembly of DHA into sperm lipids has a positive correlation with the increase in the fluidity of the membrane and, as a consequence, it positively affects the motility of the sperm cells as well [47,48,49].

The data obtained from the lipidomic analysis of the spermatozoa were correlated with the donkeys’ characteristics (BCS and age). In human studies, a high body-mass index (BMI) is linked to a lesser sperm quality [18], while the concentration of the sperm as well as the velocity seem to be irrelevant to the BMI [49]. In our study, because of having a homogeneous group of donkeys for body condition, we did not detect any differences between our fatty acid values and the BCS of the donkeys, as expected.

A correlation between the age and the levels of DHA and PUFA-ω3 content showed a significant difference, a result consistent with a study on turkeys according to which, lower levels of total PUFAs were detected in older turkeys [19]. A previous study showed that there might be a possible correlation between low levels of DHA and total PUFAs, and infertility in men [50]. Taking into consideration the fact that fertility is positively correlated with the proportion of PUFAs on the spermatozoa’s membrane [43], our data indicate a potential loss of sperm quality with age, which could potentially be the result of PUFAs’ peroxidation leading to an alteration of the membrane’s homeostatic balance [51]. Aging increases oxidative stress in semen [20,51], and this peroxidation of membrane lipids can cause a fluidity change in the cell membrane that could result in the loss of the homeostatic balance of the membrane. In this regard, it could affect the communication of the cell with its environment, having a negative influence on the signaling properties of the cell as well as the disruption of the equilibrium of the permeability of the membrane. This could lead to a decreased sperm motility, and thus, lower sperm quality.

For the first time, a correlation between fatty acid content and motility has been deduced. Our data indicate a positive correlation between stearic acid levels and motility (%). In addition, a positive correlation was also detected between EPA levels and motility (%), which is consistent with previous studies [35].

It is of high interest to mention that our data showed a positive correlation between oleic acid and progressive motility. It is reported in the literature that variations in stallion progressive motility (PM) are responsible for only 20% of the total variation in fertility, concluding that a PM lower than 40% is likely to have a negative effect on stallion fertility [21]. On the other hand, a PM higher or equal to 45% has been established as the threshold value associated with a change from lower to higher fertility [52]. A study on boar sperm demonstrated that exogenous oleic and palmitic acids resulted in increased sperm motility, PM and VSL (straight line velocity) [53]. In the same study, these two fatty acids were being used as energy substrates for ATP production via e-oxidation in the mitochondria of boar sperm. From ejaculation to fertilization, the glucose, FA and amino acids are the substrates for ATP generation. Stearoyl-CoA desaturase is responsible for the transformation of stearic acid (C18:0) to oleic acid (Cis9-18:1) in order to maintain membrane fluidity [54]. Significant differences between oleic and stearic acids that were previously found between fertile and infertile men verify the important role this transformation pathway plays in biological membrane fluidity. Another study in bulls demonstrated that sperm progressive motility is facilitated by saturated fatty acids [55]. This could be considered a paradox, since progressive motility is highly associated with fertility and the PUFA content. However, a deeper understanding of the way that fatty acid content is distributed on the spermatozoa’s membrane could provide important information about the way the fatty acids contribute to the physiological sperm parameters. It is during their transformation in the epididymis that the spermatozoa acquire their motility and, thus, their fertilizing ability [56].

It is shown in the literature that fatty acids can be differently distributed among the two different compartments of the spermatozoa and that this distribution varies among different species [43]. An example is DHA, of which the head contains a higher concentration in human sperm. [41]. A study in bulls shows that, during summer, the tail of the spermatozoa consists of a significantly higher percentage of PUFAs as well as MUFAs when compared to the head, and, specifically, oleic acid was detected in levels two times higher in the tail [57].

While it is widely accepted that PUFAs are linked to better sperm quality, the results about oleic acid are not yet conclusive, since there are contradictory studies that show a negative correlation between oleic acid and progressive motility [46].

## 5. Conclusions

This work evaluated for the first time the composition of the spermatozoa’s as well as the erythrocytes’ membranes of 26 healthy male MF donkeys with the study of a cohort of 12 fatty acids, components of said membranes, providing us with valuable information about the metabolic pathways of these fatty acids. In addition, a possible correlation between the FA structure of spermatozoa’s membranes and the donkeys’ characteristics, as well as the reproductive characteristics of the animals, offer a deeper understanding of the distribution of the fatty acid in the spermatozoa’s membranes and their role in motility, i.e., the fertility of these donkeys. However, this study has potential limitations. The Martina Franca donkey, being an endangered species, suffers from a low population, and despite the fact that our study was conducted upon a high percentage of their total population in Italy, the actual number of the selected donkeys (26) was still relatively low. On the other hand, our study is the first lipidomic assay to conduct a systematic approach in the investigation of equine sperm and/or erythrocyte lipidome, since the studies that have been conducted in the past on the subject are few, and there is a great variability on the methods used for the assessment of equine sperm, creating the necessity of a unified consistent method to assess equine sperm parameters in order to further explore the equine sector. This study could be the starting point of further investigations for a deeper understanding of the specific role of each fatty acid in the formation of the spermatozoa’s membrane, its transformation during the maturation process and its correlation with the different sperm parameters. In addition, a nutrition-based future investigation could offer additional valuable information about the dietary uptake and its reflection on the erythrocytes’ membrane composition and, through the transformation of the spermatozoa during the maturation period, it could shed light on a possible correlation between the spermatozoa and the erythrocytes.

## Figures and Tables

**Figure 1 animals-13-00008-f001:**
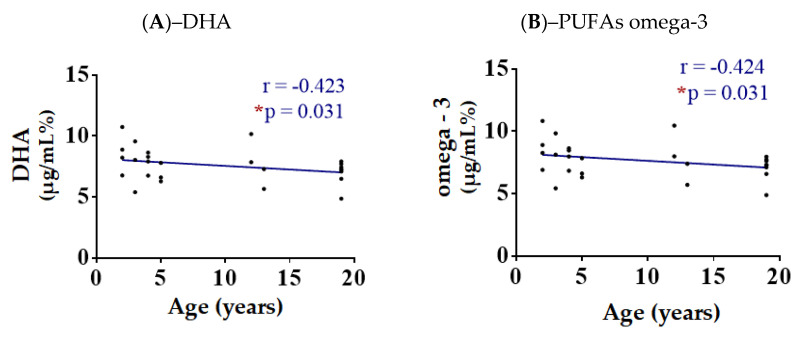
Spearman correlation with linear regression and parameters for 26 healthy donkeys using age and lipid indexes obtained from spermatozoa membranes. Values significantly different when compared to with each other: (*) *p* < 0.05. Significant decreased levels of DHA (**A**) as well as PUFAs omega-3 (**B**) with the increase in the age of the donkeys.

**Figure 2 animals-13-00008-f002:**
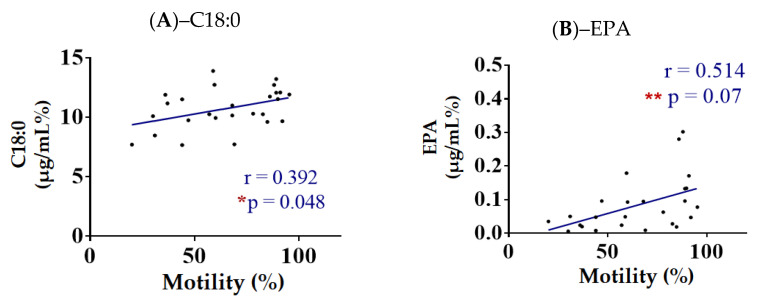
Spearman correlation with linear regression and parameters for 26 healthy donkeys (using motility (%) and lipid indexes obtained from spermatozoa membranes. Values significantly different when compared to with each other: (*) *p* < 0.05 and (**) *p* < 0.005. Significant increased levels of stearic-C18:0 (**A**) and EPA (**B**) with the increase in the motility (%) of the spermatozoa.

**Figure 3 animals-13-00008-f003:**
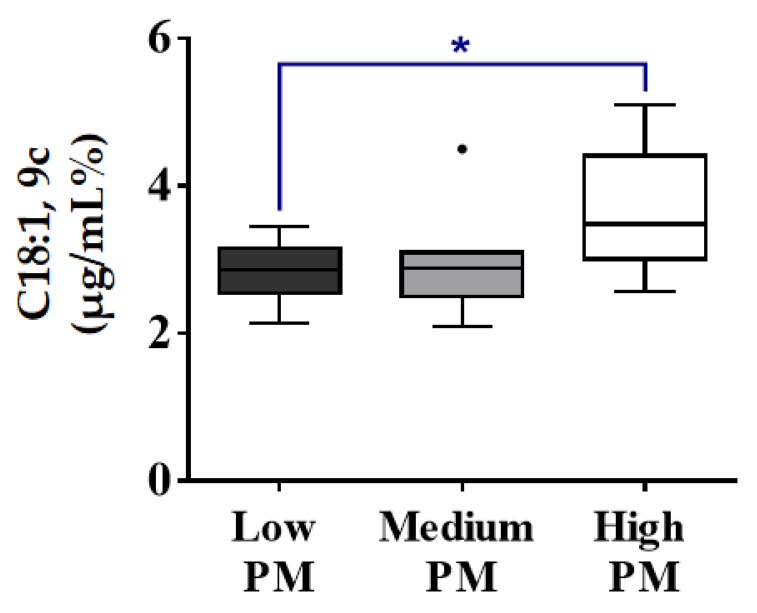
Relative quantitative percentage differences between healthy donkeys with spermatozoa that demonstrated low progressive motility % (Dark Gray, low PM < 35%, number = 9); medium progressive motility % (Light Gray, 35% ≤ medium PM < 45%, number = 7) and high progressive motility % (White, PM < 45%, number = 10) for oleic acid −C18:1,9c in the spermatozoa membranes. The values are given as mean ± SD. Values significantly different when compared to with each other: (*) *p* < 0.05. Significantly increased levels of oleic acid detected on the spermatozoa that demonstrated high PM.

**Table 1 animals-13-00008-t001:** Chemical Composition of Concentrate and Hay (% on a DM Basis) Used During the Trial.

Chemical Composition	Concentrate	Hay
DM (%)	89.73	91.47
CP (%)	18.25	13.90
NDF (%)	25.84	55.10
ADF (%)	9.45	41.16
ADL (%)	2.75	9.18
ASH (%)	7.40	7.40

DM—dry matter; CP—crude protein; NDF—neutral detergent fiber; ADF—acid detergent fiber; ADL—acid detergent lignin.

**Table 2 animals-13-00008-t002:** Sperm Parameters of 26 Healthy Donkeys.

Item	Median	Min	Max	Mean	SD
Reaction time (min)	10.0	4.0	45.0	12.6	10.1
Total volume (mL)	50.0	20.0	70.0	51.5	12.9
Gel-free volume (mL)	43.8	10.0	60.0	41.4	13.7
Concentration (mln/mL)	440.0	120.0	1600.0	517.6	379.8
Dead (mln/mL)	85.5	19.0	196.8	88.4	41.5
Motility (%)	68.0	20.0	95.3	65.2	23.1
Progressive Motility (%)	40.2	4.0	59.9	37.8	16.8

**Table 3 animals-13-00008-t003:** Cohort of 12 Fatty Acids Expressed as Percentages of the Found μg/mL Quantities Detected by Gas Chromatographic Analyses of the Fatty Acid Methyl Esters (FAMEs) After Isolation and Workup of Spermatozoa Membrane Glycerophospholipids of 26 Healthy Donkeys.

FAME (%μg/mL) *	Median	Min	Max	Mean	SD
C14:0	4.3	2.1	5.5	4.1	1.0
C16:0	30.3	24.5	35.5	30.3	2.9
C16:1	0.08	0.04	0.34	0.1	0.1
C18:0	10.7	7.7	13.9	10.8	1.7
9c,C18:1	3	2.1	5.1	3.2	0.8
11c,C18:1	3.3	1.5	6.6	3.5	0.9
C18:2	4.5	3.4	7.4	4.7	1.0
C20:3	1.95	0.8	3.1	2.0	0.5
C20:4	6.3	3.1	9.9	6.6	1.4
C20:5	0.06	0.01	0.3	0.1	0.1
C22:5	27.8	21.5	31	27.6	2.6
C22:6	7.6	4.8	10.7	7.6	1.4
SFA ^1^	45	37.9	52.4	45.1	3.8
MUFA ^2^	6.8	5.5	9.1	6.8	0.8
PUFA omega-3 ^3^	7.7	4.8	10.8	7.7	1.4
PUFA omega-6 ^4^	41	47	28.9	40.4	3.8
PUFA ^5^	48.4	41	54.4	48.0	3.7
SFA/MUFA	6.6	4.6	8.4	6.7	1.1
Omega-6/Omega-3	5.4	3.3	8.7	5.5	1.3
PUFA balance ^8^	15.76	10.3	23.1	16.0	3.0
UI ^9^	233.8	190	266	229.9	17.7
PI ^10^	261.7	209	304	258.9	20.9

* The values are expressed as percentage of the found quantities (calculated as μg/mL) ± standard deviation (SD) obtained from the gas chromatographic analyses, using calibration and quantitation protocols and standard reference compounds for each FAME. The GC peak areas of the 12 fatty acids cohort corresponds to ca. 97% of the total peak areas of the chromatogram. ^1^ SFA (Saturated Fatty Acids) = %C14:0 + %C16:0 + %C18:0. ^2^ MUFA (Monounsaturated Fatty Acids) = %C16:1 + %9c,C18:1 + %11c,C18:1. ^3^ PUFA (Polyunsaturated Fatty Acids) omega-3 = %C20:5 + %C22:6. ^4^ PUFA (Polyunsaturated Fatty Acids) omega-6 = %C18:2 + %C20:3 + %C20:4 + %C22:5. ^5^ PUFA = %C18:2 + %C20:3 + %C20:4 + %C22:5 + %C20:5 + %C22:6. ^8^ PUFA balance = [(%C20:5 + %C22:6.) / Total PUFA] × 100. ^9^ UI (Unsaturation Index) = (%MUFA × 1) + (%18:2 × 2) + (%20:3 × 3) + (%20:4 × 4) + (%20:5× 5) + (%22:5 × 5) + (%22:6 × 6). ^10^ PI (Peroxidation Index) = (%MUFA × 0.025) + (%18:2 × 1) + (%20:3 × 2) + (%20:4 × 4) + (%20:5 × 6) + (%22:5 × 6) + (%22:6 × 8).

**Table 4 animals-13-00008-t004:** Cohort of 12 Fatty Acids Expressed as Percentages of the Found μg/mL Quantities Detected by Gas Chromatographic Analyses of the Fatty Acid Methyl Esters (FAMEs) after Isolation and Workup Erythrocyte Membrane Glycerophospholipids of 26 Healthy Donkeys.

FAME (%μg/mL)	Min	Max	Mean	SD
C14:0	0.0	0.0	0.0	0.0
C16:0	8.3	12	10.3	1.2
C16:1	0.6	1.6	1.1	0.3
C18:0	7.4	12.5	9.4	1.3
9c,C18:1	27.6	33.9	30.3	1.6
11c,C18:1	0.39	1.4	0.9	0.3
C18:2	40.3	49.3	46.3	2,1
C20:3	0.09	0.17	0.1	0.02
C20:4	0.9	1.9	1.4	0.3
C20:5	0.17	0.19	0.2	0.01
C22:5	0.0	0.0	0.0	0.0
C22:6	0.01	0.09	0.03	0.02
SFA ^1^	17.9	22.5	19.7	1.0
MUFA ^2^	30	35.5	32.3	1.3
PUFA omega-3 ^3^	0.19	0.27	0.2	0.02
PUFA omega-6 ^4^	41.7	50.6	47.8	2.0
PUFA ^5^	42	50.9	48.0	2.0
SFA/MUFA ^6^	0.57	0.68	0.6	0.03
Omega-6/omega-3 ^7^	185.8	257.5	222.6	21.2
PUFA balance ^8^	0.39	0.54	0.5	0.04
(UI) ^9^	122	137	131.9	3.1
(PI) ^10^	48	58.2	54.2	2.3

^1^ Total SFA (Saturated Fatty Acids) = %C14:0 + %C16:0 + %C18:0. ^2^ Total MUFA (Monounsaturated Fatty Acids) = %C16:1 + %9c,C18:1 + %11c,C18:1. ^3^ PUFA (Polyunsaturated Fatty Acids) omega-3 = %C20:5 + %C22:6. ^4^ PUFA (Polyunsaturated Fatty Acids) omega-6 = %C18:2 + %C20:3 + %C20:4 + %C22:5. ^5^ Total PUFA = %C18:2 + %C20:3 + %C20:4 + %C22:5 + %C20:5 + %C22:6. ^6^ SFA/MUFA = (%C14:0 + %C16:0 + %C18:0)/(%C16:1 + %9c,C18:1 + %11c,C18:1). ^7^ Omega-6/omega-3 ratio = (%C18:2 + %C20:3 + %C20:4 + %C22:5)/(%C20:5 + %C22:6.). ^8^ PUFA balance = [(%C20:5 + %C22:6.)/Total PUFA] × 100. ^9^ UI (Unsaturation Index) = (%MUFA × 1) + (%18:2 × 2) + (%20:3 × 3) + (%20:4 × 4) + (%20:5× 5) + (%22:5 × 5) + (%22:6 × 6). ^10^ PI (Peroxidation Index) = (%MUFA × 0.025) + (%18:2 × 1) + (%20:3 × 2) + (%20:4 × 4) + (%20:5 × 6) + (%22:5 × 6) + (%22:6 × 8).

## Data Availability

The data presented in this study are available on request from the corresponding author.

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
