# Peer review of "The Lipidomics of Spermatozoa and Red Blood Cells Membrane Profile of Martina Franca Donkey: Preliminary Evaluation"

_animals, 2022, doi:10.3390/ani13010008_

Round 1

Reviewer 1 Report

Dear authors, 

thank you for this piece of work. I found it meritorious of being considered further to publication, though in my opinion the structure of the manuscript needs to be improved at this stage.

In view of the topic, I realized that the title and the abstract are not consistent with the Introduction. Authors start with the description of one single breed of donkeys. This is not expected, for expectations raised in the reader are different. Please amend this aspect. I believe that the paper must start from the lipidomic description and physiologic/diagnostic/dietary or nutritional importance. Moreover, the introduction is not well developed and some citations turned out to be out of context (likewise in the discussion). Indeed, if considering lipidomic Sassot et al., Equine Vet J 2019 Sep;51(5):696-700.doi: 10.1111/evj.13064, should be a reference. More recently, Goodrich and Behling-Kelly Animals 202212(20), 2746; https://doi.org/10.3390/ani12202746 , have explored into details from a general context, what you specifically describe in this case as to sperm cells and erythrocytes lipidomic profile in a donkey breed. At this regard, indeed, fats and vitamin E were widely explored, with vitamin E being involved  as an integral component of erythrocytes membrane, and for which intestinal absorption is enhanced in the presence of dietary fats. I would invite authors to refer to Vatassery (1989), 24, 209-304 Lipids https://doi.org/10.007/BF02535167. In addition to this, the vitamin E status in humans and some animal species is referred to the proportion of circulating cholesterol and triglycerides. Nowhere these aspects are cited in the text, which are instead fundamentals of the background. It is a belief of this reviewer that such information should be reported in the introduction. Moreover, the donkey breed should be considered as a case, not even as a model of preliminary results proving to assume further general rules from 26 donkeys of one single breed. So, in general, some argumentations might benefit of toning down a bit or change the approach.

Comparatively speaking, therefore, the reference to acquaintances in this regard available for the horses appears to be necessary, albeit aspects about donkey (very different according to breeds) are also reported. As a starting point, I would suggest that the manuscript might increase its soundenss  if its structure is substantially changed, in relation to the order of descriptions provided in the introduction and discussion. The attempt of authors to avoid useless citations to papers which appeared to me out of context, would be very appreciated. Rather, sharpen the core of the investigation to the relevant references. In view of this, also the rationale behind the investigation appears to be somewhat hazy, the way it is reported/unreported. Thus, the hypothesis does not descend from appropriate description of the actual knwoledge, limiting the gap to the lack of description, simply! Why is it important? I found a sort of mixing up with oxidative stress, membrane stability and seasonality. Maybe all of them, but expressed much better... maybe! On simply googling the effect of vitamin E, (and not only of phospholipids), on erythrocytes of breeding stallions and semen quality you find Cappai et al., 2021, Biol Trace Elem Res 199, 3287–3296 (2021). https://doi.org/10.1007/s12011-020-02447-7 worthy of being cited too, to depict the state of the art, among others which are not used to the purpose, unfortunately.

The Material and Methods are adequately described and can allow for the repetition of the experiment.

The discussion suffers from the same, though easily amendable, limits expressed about the structure and content of the introduction. I would invite authors to render the manuscript more focused on the core results, discuss necessary reference and not argue with out-of-context aspects, which may lead far from the objectives of the the experiment, which risk to leave the exptectations of the reader unmet.

As a consequence, conclusion should be tailored to main findings.

Thank you.

Author Response

We would like to thank the Reviewer for the attention to the improvement of our paper entitled “The lipidomics of spermatozoa and red blood cells membrane profile of Martina Franca Donkey: preliminary evaluation“ to evaluate the possibility for publication on  “Animals” - Special Issue “Advances in Equine Metabolomics

We responded point by point to the comments received by the Reviewer and attached our replies and the new version of the manuscript.

Thanking you in advance for your attention

Reviewer 2 Report

The manuscript studies the correlation between the fatty acid profile of spermatozoa and erythrocytes with sperm quality in Martina Franca donkey. 

The study is interesting because of the scarce literature on the subject, especially in donkey, but there are important aspects of the experimental design that should be clarified by the authors and which, if not well justified, could invalidate the publication of this work.

My main criticism refers to the number of samples obtained per animal. You are working with 26 donkeys but how many samples per donkey?, if it is only 1, I consider that the results should be taken with great caution. In the case of several samples per male were obtained, it would be more appropriate but then the statistical analysis would have to include the random effect of the male. Statistical analyses should be repeated with the improved model.

I believe that the results presented on the correlation between AF and age (3.3.1) are not correct. If you have only 1 data per animal, the effect of age and male are statistically confounded and no conclusions can be drawn.

Other considerations:

. Title: "healthy Martina Franca Donkey". The authors do not carry out any objective verification of the health status of the animals.

. the general objective (lines 104-106) is not appropriate. No comparison between animals with different health status is made.

. statistical analysis: see my general comment.

. Table 1. It would be more convenient to present it with a structure similar to table 2, i.e. mean, range, sd and coefficient of variation.

. Results: in general, the data presented in the text do not correspond to those presented in the tables (please use the same number of decimals in table and text).

. Table 2 and fig. 1 is presented up to 3 times in the text (line 240, 261-262, 275-276). In addition figure 1 provides repeated information in table 2. Please rewrite the presentation and consider deleting figure 1.

. 3.4.2 Progressive motilty. In my opinion the information concerning the classification of PM into low and high should be indicated in the material and methods. Furthermore there is a group of samples between 35 and 45% PM that are not considered in the analysis, an intermediate group could be proposed and analysed.

. Discussion: lines 451-461. please reconsider taking into account the suggestions.

. Lines 445-449. All animals had similar body condition, this parameter has not been studied by the authors and is therefore out of place for discussion. Line 445-446: this sentence is not correct.

The conclusions do not meet the objectives. Moreover, they are over-conclusive (line 515-516) as no studies on stress or welfare have been carried out in this work.

Minor remarks:

Line 24: "The"

Line 30: remove "donkey behaviuor/welfare"

Line 67: delete ","

Figures 2 and 3: Remove "using the data of Tables 1 and 2"

Figure 3: Uniformize "Motili" or "Motility"

Line 414, 441: remove quotes

T

Author Response

We would like to thank the Reviewer for the attention to the improvement of our paper entitled “The lipidomics of spermatozoa and red blood cells membrane profile of Martina Franca Donkey: preliminary evaluation“ to evaluate the possibility for publication on  “Animals” - Special Issue “Advances in Equine Metabolomics

We responded point by point to the comments received by the reviewers and attached our replies and the new version of the manuscript.

Thanking you in advance for your attention

Round 2

Reviewer 2 Report

Authors have considedered properly all suggesions and comments.

Congratulations